# Considering Two Aspects of Fish Welfare on African Catfish (*Clarias gariepinus*) Fillet throughout Postmortem Condition: Efficiency and Mechanisms

**DOI:** 10.3390/foods11244090

**Published:** 2022-12-17

**Authors:** Nima Hematyar, Aiman Imentai, Jiří Křišťan, Swapnil Gorakh Waghmare, Tomáš Policar

**Affiliations:** 1Faculty of Fisheries and Protection of Waters, South Bohemian Research Center of Aquaculture and Biodiversity of Hydrocenoses, Research Institute of Fish Culture and Hydrobiology, University of South Bohemia in Ceske Budejovice, Zátiší 728/II, 389 25 Vodňany, Czech Republic; 2Faculty of Natural Sciences, Department of Ecology, Comenius University in Bratislava, Mlynská Dolina, Ilkovičova 6, 842 15 Bratislava 4, Slovakia

**Keywords:** fish welfare, oxidation progress, stock density, bled, unbled

## Abstract

Knowledge about fish welfare and its impact on fish fillet quality is still insufficient. Therefore, the influence of two aspects of fish welfare (slaughtering method: bled and unbled fish; fish stock densities: 90, 120, and 150 kg·m^−3^) on African catfish fillet quality during postmortem conditions was investigated. The aim of study was to determine (i) the efficiency of bleeding on oxidation progress and (ii) the influence of stock density on fillet quality. Sodium dodecyl sulfate-polyacrylamide gel electrophoresis (SDS–PAGE) showed a higher protein loss in the unbled than in the bled groups, especially in the heavy myosin chain (MHC) band. However, density did not show any influence on protein profile. Western blot analysis showed fewer oxidized carbonyls in the bled than in the unbled groups; higher oxidation development, microbial growth, and lower hardness were observed in unbled fillets. Additionally, hardness was higher at 90 and 120 kg·m^−3^ densities in bled fillet compared to 150 kg·m^−3^. The first three days of storage showed a higher oxidation rate in unbled fillets than in bled fillets, confirming the contribution of hemoglobin to oxidation development with different mechanisms of protein oxidation. The obtained results revealed the same fillet quality in all aspects at either 90 or 120 (kg·m^−3^) stock densities, which would suggest 120 kg·m^−3^ for the fishery industry. However, higher stocking density in this study would not be appropriate for fish welfare.

## 1. Introduction

African catfish (*Clarias gariepinus*) are intensively cultured, mainly in the African continent as well as the Middle East and Asia [1]. The FAO reported that global African catfish production was five million tons during 2021 [2]. Due to the higher growth rate of Clarias gariepinus rather than other Clarias species, it has been introduced and cultured outside of its geographical range. Furthermore, compared to other fish species, this catfish species is more resistant to inconvenient environmental conditions, and it can breathe air because of its particular gill morphology [3]. African catfish fillets are well-known for their fillet quality and flavor, making them suitable for processing [4].

Fish fillets contain heme proteins (hemoglobin and myoglobin), which are responsible for developing lipid oxidation as well as discoloration [5]. Metal catalysis is an important lipid oxidation reaction in fish fillets after post-slaughter processes, which can start the autoxidation progress [6]. Bleeding is an important process that can be investigated as a slaughtering step. Due to the bleeding, iron as a source of lipid oxidation from the fish body is partially eliminated. Additionally, immediate bleeding preserves the freshness of fillets for 15 days of storage [7]. On the other hand, inadequately bled or non-bled fish had lower overall quality [8]. In addition, to date, the quality of catfish fillets by the effect of bleeding has not been studied.

Endogenous enzyme activity followed by protein degradation and microbial growth has a crucial impact on seafood product quality [9]. The quality of seafood products deteriorated due to autolytic changes and oxidation development [10]. Particularly, oxidation of myofibrillar proteins can negatively influence on water holding capacity and hardness [11]. It is assumed that the aggregation of actin and myosin during storage is the primary reaction to muscle food’s decline in hardness and water holding capacity (WHC) [12]. Considering fish welfare would be difficult due to the lack of agreement about the feeling of pain and anxiety in fish, and finding an objective methodology to investigate the correlation between fish welfare and fillet quality is also difficult [13]. Additionally, aquaculture conditions such as stock density, transportation, water quality and water temperature have an effect on fish welfare and can also affect fish fillet quality [14,15]. On the one hand, increasing stock density enhances productivity, and on the other hand, high densities are associated with lower fish welfare conditions. Moreover, mapping a relationship between stock density and fillet quality is needed for economical fish rearing conditions to achieve better final products.

This study investigated (i) the efficiency of bleeding and the role of hemoglobin on lipid oxidation progress; (ii) the protein profile of fish fillets reared under different stock densities; (iii) the consequences of fish welfare affected by fish density and slaughtering methods on several fillet quality parameters with respect to lipid and protein oxidation progress, autolytic and textural changes, as well as calpain activity during postmortem. The hypotheses of the current study were as follows: (i) that the density and slaughtering methods might affect the quality of fish muscle upon postmortem time; (ii) oxidative changes as well as other quality factors were interactive. The obtained results indicate the role of density on the fish fillet quality and may also aid in the enhancement and development of fish production and fillet quality.

## 2. Materials and Methods

### 2.1. Experimental Design

All animal protocols and procedures were approved by the University of South Bohemia Research Institute of Fish Culture and Hydrobiology and Use Committee (No. 16OZ19179/2016-17214). Sixteen-month-old marketable African catfish (N = 342; body weight 2395.60 ± 287.02 g) were cultured under three different constant fish stock densities (90, 120 and 150 kg·m^−3^) for three months in the RAS system of the Happy Fish tilapia fish farm from the Czech Republic. These catfish were rested without any feeding for 24 h in a water tank (water temperature: 26 °C and oxygen saturation: 96%) before the start of the experiments. Regarding this experiment, in total, nine tanks with volume 1750 L were used for culture which means each density group had three replicates and each tank cultured with 38 fish.

Thirty-six randomly selected catfish from each stock density (the number of fish was the same in all groups with respect to the density and slaughtering methods) were bled by cutting the gill arches and keeping them straight on ice for 60 min to prevent coagulation and improve bleeding. The unbled fish (*n* = 36) (control) from each stock density were killed by percussion by an expert person and kept on ice.

In this study, the first group (bled) was sacrificed with a blow to the head, followed by cutting the gills. The second group (unbled) was sacrificed with a blow to the head. After that, both groups were kept for 1 h on ice so both groups were in the same situation until further processing. Moreover, we caught all fish from both groups (bled and unbled) in the same condition and with an expert person to minimize the physical activity of fish and reduce the stressful situation.

After slaughtering, both unbled and bled fish from each density were washed with cold water and filleted (two fillets from each fish). The fillets from both samples were packed individually in plastic bags (size 250 mm × 400 mm) to avoid the drying of the surface of the fillets and kept in a straight position in a refrigerated chamber at 4 °C. The assessment of fillet quality was carried out immediately after filleting and later every 3 days, after 0, 3, 6 and 9 days of refrigerator storage. The fillets from each group were homogenized and stored at –80 °C until analysis. Six randomly chosen left fillets from each group (considering slaughtering methods and density) were used for liquid loss and hardness analyses, and six randomly right fillets (due to the six replications for each analysis at each time point we used the same fillet) were used for TBARS, protein carbonyl, proteomic, calpain and hemoglobin as chemical analyses (Appendix A).

Additionally, in aggregate, 270 fish (whole fish) were used to consider pH and rigor index. For this purpose, five fish from each group (with respect to density and slaughtering methods) were randomly separated and kept on ice, the rigor index and pH were monitored at the specified interval times (0, 4, 6, 10, 20, 24, 48, 72 and 96 h).

### 2.2. Rigor Index

The rigor mortis development was measured by the Cuttingers method (tail drop) [16]) on five fish from each group (considering density and slaughtering method) during 96 h post-slaughtering. The fillets were kept on ice during the rigor index analyses. The rigor index (Ir) was calculated by the formula Ir = [(Lo − Lt)/Lo] × 100, where L represents the vertical drop (cm) of the tail when half of the fish fork length is placed on the edge of a table as a function of time. The tail drop at the beginning of the experiment is Lo, while Lt represents measurements throughout the experiment (t = 0 − 96 h). T = 0 means immediately after slaughtering.

### 2.3. PH

The pH (*n* = 5 fish) from bled and unbled and each density was measured by inserting a pH probe (Testo 206, Lenzkirch, Germany) into the upper mass of the fillet, just behind the head.

### 2.4. Total Viable Counts and Psychrophilic Bacterial Counts

The total viable counts (TVC) and psychrophilic bacterial counts (PBC) were determined in plate count agar according to the Chinese National Standard of 2008 (GBT 4789.2–2008). A fillet sample (10 g) from bled and unbled fillets (*n* = 5) was taken and transferred into a stomacher bag containing 90 mL of 8.5 g/L sterile NaCl water and was then homogenized for 1 min. A serial 10-fold dilution was performed, and then 1 mL of each dilution was pipetted onto the surface of plate count agar (Base Bio Tech, Hangzhou, China) plates, which were incubated for 72 h at 30 °C and 7 °C for 7 days for TVC and PBC, respectively. Microbial counts (TVC and PBC) were expressed as log cfu/g.

### 2.5. Calpain Activity Assay

Calpain activity was measured on (*n* = 5) fillets from each group by using a calpain activity assay kit (BioVisison, Mountain view, CA, USA) (only activated calpain enclosed by the cytosol was detected by the kit) according to the published protocol [17]. Briefly, 100 µg of each type of muscle sample was homogenized in an extraction buffer (extracts cytosolic proteins and avoid contamination by lysosome proteases and cell membrane), and then the concentration was diluted to 200 µg protein in 85 µL. In total, 10 μL of reaction buffer (10×) and 5 µL of calpain substrate (Ac-LLY-AFC) were added, followed by incubation at 37 °C for 1 h. The samples were measured by a plate reader with a 400-nm excitation filter and a 505 nm emission filter. The change in calpain activity was expressed as the relative fluorescence fold change.

### 2.6. Hardness Analysis

Hardness analysis (*n* = 6 fillets from each group) was performed instrumentally by using a texture analyzer (TA-XT. Plus, Stable Micro systems Exponent- [Graph 1. (0:0)], UK). Therefore, a flat-ended cylinder (10 mm diameter, type P/10) was pressed into the section of the fillet below the dorsal fin perpendicular to the muscle fibers at a speed of 2 mm/s. The press was stopped when the fillet was compressed to 50% of its original thickness [18]. Hardness was defined as the maximum force detected during the first compression, expressed in grams.

### 2.7. Liquid Loss

It was measured on six fillets from each group with a texture analyzer (TA-XT. Plus, Stable Micro systems UK) by pressing a flat-ended cylinder (10 mm diameter, type P/10) into the fillet below the dorsal fin perpendicular to the muscle fibers at a speed of 2 mm/s until it reached 50% of the fillet height and held for 60 s. A dry, pre-weighed filter paper will be placed under the sample. The filter paper will then be weighed immediately after the test, with and without the fish piece, and the liquid loss will be calculated.

### 2.8. Heme Content

Total heme content was assessed with a slight modification according to the [19] method. A 2 g ground sample was transferred into a polypropylene centrifuge tube (50 mL) and homogenized with 20 mL of 40 mM phosphate buffer (pH 6.8) with a homogenizer (13,500 rpm for 10 s). Then, the mixture was centrifuged at 3000× *g* for 30 min at 4 °C, and Whatman no. 1 filter paper was used to filter the supernatant. The total heme content was measured by direct spectrophotometry at 525 nm by using a UV-1601 spectrophotometer. The obtained data were represented as molal concentration of hemoglobin in μmol/kg.

### 2.9. Extraction of Muscle Proteins

Pieces of 100 mg of frozen fish fillet were cut and weighed at −20 °C to minimize artifactual protein degradation. The frozen muscle tissue was homogenized in a 500 µL volume in 50 mM phosphate buffered saline (PBS) (saline solution with a phosphate buffer concentration of 0.01 M and a sodium chloride concentration of 0.154 M, pH = 7.4). Then, the homogenized sample was centrifuged, and crude extracts (sarcoplasmic protein) were transferred to an Eppendorf tube.

### 2.10. SDS–PAGE

Sodium dodecyl sulfate–polyacrylamide gel electrophoresis (SDS–PAGE) was performed according to the method of [20]. Each sample (20 µL) was mixed with Laemmli sample buffer (sodium dodecyl sulfate (SDS), thiol agent, glycerol, tris-hydroxymethyl-aminomethane (tris), bromophenol blue) with a final protein concentration of 2 µg/µL, followed by heating for 2 min at 95 °C. Afterward, the samples were loaded onto a 10% Criterion Tris glycine Gel (Bio-Rad, Hercules, CA, USA) and subjected to electrophoresis at a constant electrical potential of 200 V. The Spectra Multicolor Broad range protein ladder (15–220 kDa) (Thermo Scientific, Rockford, IL, USA) was used as a marker. After electrophoresis, the gel was stained with 0.5% Coomassie Brilliant Blue G-250 (Bull Korean Chem Soc. 2002). The image analysis of gels was performed using Image Lab software (Molecular Imager Chemi Doc XRS+, Bio-Rad Laboratories, Hercules, CA, USA).

### 2.11. Immunoblotting

For immunoblotting, the 2,4-dinitrophenyl hydrazine (DNPH) reaction was directly performed on the protein homogenate (described in Section 2.10) that was obtained by centrifugation of the protein homogenate at 12,600× *g* for 3 min. The supernatant was used for analysis.

The protein concentration was adjusted to 10 mg/mL by using a BCA kit (Pierce, Rockford, IL). Protein carbonyls were derivatized by mixing 20 µL of sample (1:2) with 12% sodium dodecyl sulfate (SDS), 10% TFA (trifluoroacetic acid), and 10 mM DNPH and were incubated for 30 min at room temperature. The reaction was stopped by adding 40 µL neutralization buffer (1:2) containing 2 M Tris-base, 30% glycerol, and 20 mM dithioerythritol (DTE) before it was separated by SDS–PAGE.

The samples were centrifuged for 3 min at 12,600× *g* and loaded onto a gel (10% Tris glycine gels: Bio-Rad, Hercules, CA, USA). SDS–PAGE was performed as described above in Section 2.3. After SDS–PAGE, the gels were placed on 0.2 µm polyvinylidene difluoride (PVDF) membranes (Bio-Rad, Laboratories, USA) and electrically transferred using a Trans-Blot SD, semidry transfer cell, 0.35 A, max 50 V for 60 min (Bio-Rad, Laboratories, USA). After transfer, the membranes were blocked with 5% skim milk in Tris-buffered) and incubated with anti-DNP produced in rabbit (Sigma Aldrich, Taufkirchen, Germany) at a 1:16000 dilution in TBS overnight at +4 °C. The membranes were washed in TBS and incubated in the secondary antibody (1:8000 dilution), which was horseradish peroxidase-conjugated swine anti-rabbit (DAKO Denmark A/S). After washing in TBS, the blot was developed using an ECL ± kit (Clarity Western Bloting Substrate) (Bio-Rad Laboratories, USA) to detect the proteins. The image analysis of gels and blots was performed using Image Lab software (Molecular Imager Chemi Doc XRS+, Bio-Rad Laboratories, USA).

### 2.12. Protein Carbonyl Quantification with 2,4-Dinitrophenylhydrazine

Protein oxidation was estimated as carbonyl content after incubation with 2,4-dinitrophenylhydrazine (DNPH) in 2 N hydrochloric acid following a slightly modified method described by (Levine et al., 1990) on six fillets from each group. The carbonyl concentration was analyzed as DNPH calculated based on the absorption of 21.0 mM^−1^ cm^−1^ at 370 nm for protein hydrazine. The protein concentration was measured at 280 nm in the same sample and quantified by using bovine serum albumin as a standard.

### 2.13. Thiobarbituric Acid Reactive Substances (TBARS)

Lipid oxidation was measured by the thiobarbituric acid reactive substances (TBARS) method according to the method of [21]. The semi-frozen samples (six fillets from each group) were minced, connective tissues and visible fat were removed, and a subsample of approximately 1 g of muscle tissue was taken for extraction. The samples were homogenized with 9.1 mL (0.61 mol/L) trichloroacetic acid (TCA) solution and 0.2 mL (0.09 mol/L) butylated hydroxytoluene (BHT) in methanol using an UltraTurrax (Janke & Kunkel, Staufen, Germany, T25IKA-Labortechnik) for 3 × 20 s at a speed of approximately 14,000 rpm. Afterward, the homogenate was filtered through Munktell paper (Munktell Filter AB, Grycksbo, Sweden). Two times, 1.5 mL of the filtrate was transferred to new tubes, and 1.5 mL of thiobarbituric acid (TBA) solution (0.02 mol/L) or water was added to the first (test sample) and second (sample blank) samples, respectively. After the reaction proceeded in darkness for 15–20 h (overnight) at room temperature (20 °C), the reaction complex was detected at a wavelength of 530 nm against the sample blank using a UV–visual spectrophotometer (Specord 210; Analytik Jena, Jena, Germany). The amount of TBARS was expressed as malondialdehyde (MDA) (µg/g).

### 2.14. Statistical Analysis

The statistical evaluation was performed by using GraphPad Prism 9.4.1 (GraphPad Software, San Diego, CA, USA). Three-way ANOVA followed by Tukey’s multiple com-parisons test was used to determine the differences in the storage time (0, 3, 6 and 9 days) in bled and unbled groups at different stock densities (90, 120 and 150 kg·m^−3^). The microbial data were analyzed using two-way ANOVA analysis in the Statistica CZ 12 software package. The postmortem storage efficiencies on rigor index, pH, calpain activity, liquid loss, heme content and lipid and protein oxidation were also compared among factors using a three-way ANOVA (R- statistical programme, version 4.2.2) with stocking densities (90, 120 and 150 kg·m^−3^), slaughtering methods (bled and unbled) and storage time (0, 3, 6 and 9 days). The normality and homogeneity of the data was carried out before ANOVA analysis. Differences were considered significant at *p* < 0.05, and the results are presented as mean ± S.D.). The level of significance was considered at (*p* < 0.05) and the results are presented as mean ± S.D.) A three-way ANOVA *p*-value table for the data variables is included in Appendix A.

## 3. Results

### 3.1. Rigor Index

The results of the rigor index after 96 h post fish killing kept at +4 °C are shown in Appendix A. Generally, the onset of rigor is faster in unbled fish fillets than in bled ones (*p* < 0.0001). The rigor mortis was started after 6 and 10 h in unbled and bled fish, respectively, except bled fish reared at 150 kg·m^−3^ (*p* < 0.0001). Moreover, stock density could affect the onset of rigor mortis in this study (*p*< 0.001), especially for bled fish, on a time between 4 to 24 h (*p* < 0.0001). However, we observed more or less the same rigor pattern in bled fish at 90 and 120 kg·m^−3^ (*p* < 0.055) and significant differences with 150 kg·m^−3^. On the other hand, we could not observe the affect of density on unbled fish rigor index (*p* < 0.055). Additionally, the rate of rigor mortis was higher in unbled fish than in bled fish. Moreover, we observed a significant difference between the time to reach the maximum index in bled and unbled fillets (*p* < 0.0001). Full rigor in the unbled and bled fillets was observed after 20 and 24 h, respectively, except the bled fish fillet, was stocked at 150 kg·m^−3^.

### 3.2. PH

The pH results from each treatment are shown in Appendix A. Generally, we observed a significantly higher pH in bled fillets than in unbled fillets regardless of fish stock density (*p* < 0.001). Moreover, bled fish indicated a slower pH decline than unbled fish. Furthermore, pH decreased markedly by elapsing time in both bled and unbled groups (*p* < 0.0001). Stock density affected pH in some timepoint of bled and unbled fish during 96 h of refrigerator storage (*p* < 0.0001).

### 3.3. Psychrophilic Bacterial Count (PBC) Total Viable Count (TVC)

The psychrophilic bacterial count (PBC) and total viable count (TVC) results are shown in Table 1. The present results revealed that bleeding and storage significantly affected the microbial growth of African catfish fillets. We observed an increasing trend of microbial growth in raw African catfish fillets during storage (*p* < 0.05), while bled fillets showed lower microbial growth than unbled fillets (*p* < 0.05).

The TVCs were 2.2 and 2.3 log CFU g^−1^ in the fresh bled and unbled fillets, respectively (Day 0), and then increased significantly (*p* < 0.05) to 5.4 and 6.7 log CFU g^−1^ after 9 days of storage (Table 1). The PBC from both bled and unbled fillets increased significantly (*p* < 0.05) from 2.5 log CFU g^−1^ to 6.6 and 7.4 log CFU g^−1^, respectively.

### 3.4. Calpain Activity

The calpain enzyme activity decreased significantly (*p* < 0.05) during refrigerator storage in all bled and unbled groups (*p* < 0.8550) (Figure 1). Generally, unbled fillets showed higher calpain activity (not statistically significant) than bled fillets in all densities. Additionally, stock density did not affect calpain enzyme activity at 0 and 9 days in bled samples; (*p* < 0.6870) however, significant differences were detected at times 3 and 6 days between 90 and 150 (kg·m^−3^) stock density (*p* < 0.001). Furthermore, at time 0 in unbled samples, a significant difference was observed in calpain activity between 120 and 150 (kg·m^−3^) stock density. However, we could not observe any differences in calpain activity of unbled fillets at times 3, 6, 9 days storage among all densities.

### 3.5. Hardness

The hardness was significantly (*p* < 0.05) higher in bled samples than in unbled samples during refrigerator storage. Furthermore, hardness decreased drastically (*p* < 0.01) during storage time in all groups. However, we observed a significant increase on the last day of storage in unbled groups at 120 and 150 (kg·m^−3^) stock density (Figure 2). In addition, the stock density revealed a significant difference between the hardness of fillets. Among the samples, 90 and 120 (kg·m^−3^) stock density in bled and unbled fillets showed higher hardness compared to 150 (kg·m^−3^) stock density but not significantly (*p* < 0.41). Moreover, after 6 days of storage, unbled fillets revealed markedly higher hardness in 90 compared to 120 (kg·m^−3^) stock density (*p* < 0.05). The major reduction in hardness was observed after 3 days of storage in all treatments, whereas we observed less hardness reduction over the storage time.

### 3.6. Liquid Loss

We observed a significant (*p* < 0.092) increase in the amount of liquid loss in bled and unbled groups by elapsing time. Considering the effect of different stock densities over time in both groups did not show any impact on liquid loss (Figure 3). Additionally, at 90 and 150 (kg·m^−3^) stock density, we could not observe any significant differences in liquid loss between bled and unbled fillets, but markedly higher liquid loss detected after 3 days of storage at 120 (kg·m^−3^) stock density.

### 3.7. Heme Content

The total heme (Hb) content changes from the bled and unbled samples during refrigerator storage are depicted in Figure 4. Significantly (*p* < 0.0001) lower Hb was observed in all bled samples in comparison to unbled samples during storage time. The lower Hb at day 0 in bled samples compared to unbled samples indicated the efficiency of bleeding in the current study. Furthermore, Hb decreased by elapsing time by considering slaughtering methods and different densities (*p* < 0.232). Additionally, Hb decreased by elapsing time (*p* < 0.0001) by almost 50% in all groups at the end of storage time. Moreover, we detected a significantly (*p* < 0.001) lower amount of Hb at 90 and 120 (kg·m^−3^) than 150 (kg·m^−3^) stock density in all bled and unbled groups.

### 3.8. Lipid and Protein Oxidation

Regarding the lipid and protein oxidation progress, the amount of MDA (*p* < 0.001) and protein carbonyls (*p* < 0.0001) increased significantly throughout the storage period; however, both examined parameters were higher in unbled fillets than in bled fillets (Figure 5 and Figure 6). Nevertheless, stock density revealed no impact on lipid (*p* < 0.703) and protein (*p* < 0.418) oxidation development in all bled and unbled groups.

The amount of MDA and protein content increased almost three times (240%) and two times (110%), respectively, in unbled fillets after 3 days of storage; however, bled fillets indicated almost 50% and 24% increases, respectively, during the same time point. After that, the oxidation rate significantly dropped to 35% and 22%, respectively, in unbled fillets, but for the bled fillets, it did not change.

### 3.9. Proteomic Changes during Postmortem

The protein profile of African catfish fillets revealed several bands from 15 to 220 kDa Figure 7a and Figure 8a. The bands identified could be attributed to myosin heavy chain (MHC) at 200 kDa, nebulin (107 kDa), actin (43 kDa), troponin (30 kDa), and myosin light chain components (25–15 kDa) [22].

A comparison among the protein profiles of all groups (with respect to density and slaughtering methods) indicated a similar protein pattern. SDS–PAGE showed that the intensity of some protein bands decreased or fainted over time, mainly MHC. Additionally, we observed more fainted bands in the unbled groups than in the bled groups, especially in the MHC band.

Western blot (immunoblot) results Figure 7b and Figure 8b of protein carbonyl groups revealed that the band intensity increased in all groups during postmortem conditions, approximately 15–25 kDa, confirming protein degradation in MLC. On the other hand, we observed less oxidized carbonyl in bled groups compared to unbled groups.

## 4. Discussion

The consequences of fish welfare and its impact on the fish farm production system and fish fillet quality are crucial to consider. In the current study, rigor mortis was started after 6 and 10 h in unbled and bled fish, respectively, which means rigor mortis was delayed by bleeding. Similar results have also been reported by [23,24], who found that unbled fish go to rigor faster than bled fish. The delay of rigor onset in bled fish compared to unbled fish could be related to the outflow of a collagenolytic enzyme that exists in the blood [25]. Collagen is considered the most productive aggregating agent [26]. Therefore, direct gutting leads to quick vasoconstriction of the capillaries in the muscle and initiates spontaneous blood coagulation. Consequently, bleeding followed by gutting enhances the outflow of the collagenolytic enzyme from fish muscle and postpones rigor onset.

Moreover, the rigor index in bled fish was affected by stock density, while we could not observe this impact in unbled fish. In this regard, in bled fish stocked at 150 kg·m^−3^, rigor mortis was started earlier compared to 90 and 120 kg·m^−3^ in unbled fish, revealing the importance of density on the onset of rigor mortis rather than bleeding. Most likely, fish are exhausted at higher densities, and even bleeding could not postpone rigor onset.

Ref. [27] indicated some physical and physiological stress according to the procedures. Preslaughter reactions in the animal can significantly affect the final muscle quality by changing the muscle energy metabolism. For example, fatigue is related to lower muscle energy and, consequently, lower early postmortem pH [28]. The informal observations during the slaughter indicated more movement in unbled fish than in bled fish after percussion. The outcome of this activity leads to lower early postmortem pH at the beginning as well as earlier rigor mortis in unbled fish [29]. On the other hand, higher stock density is associated with greater stress during rearing, potentially leading to lower muscle glycogen levels, which may lead to higher ultimate pH [30]. This is coherent with the higher pH observed 48 and 72 h postmortem in the bled fish. Furthermore, we observed an increasing trend in all treatments after 48 h due to higher microbial activity [31]. The TVC values of bled and unbled fillets did not exceed 7 log CFU g^−1^ throughout 9 days of storage, which is the upper acceptable limit of TVC for fish fillets (7 log CFU g^−1^ is assumed spoiled) (ICMSF 1986). TVC indicated that the bled fillets were of good quality (fillets showed low microbial growth) and that the unbled fillets were in an acceptable condition (fillets were safe to consume but they were close to being spoiled). However, unbled fillets reached 6.7 log CFU g^−1^, which is close to 7 log CFU g^−1^. Lower TVC and PBC in bled fillets in comparison to unbled fillets revealed the efficiency of bleeding in retarding bacterial growth by devoiding the available nutrients in fillets [7]. These results are in accordance with earlier findings by [7] and [8], who reported a significant effect of bleeding on microbial growth in Asian seabass (*Lates calcarifer*) and common carp (*Cyprinus carpio*) fillets during storage. The presence of blood in unbleed fillets can provide a proper substrate for microorganism growth, evidenced by the higher amount of TVC and PBC throughout the 9-day storage time [7]. Therefore, with respect to the microbiological point, we suggest 6 and 9 days of shelf life for unbled and bled African catfish fillets during refrigerated storage. Additionally, PBC showed a higher number than TVC in both fillets after 6 days, confirming that psychrophilic bacteria were dominant and prevented mesophilic bacterial development.

The decreasing trend of calpain activity in all of the samples over time in the present study is in accordance with previous findings [32,33]. Calpain autolyzes during the proteolytic process and levels decrease therefore concomitantly [34]. The lower calpain activity in fish raised at lower compared to higher stocking density at days 3 and 6, is indicative of a greater initial rate of autolysis (greater proteolytic activity) of calpain in fish raised at lower density. Higher stocking densities are not only likely to cause chronic stress but also to increase stress reactivity, potentially leading to stronger stress reactions during slaughter. These observations are in agreement with results in other species, indicating that the tenderization process is altered in stressed animals and in animals with greater stress reactivity [29]. Calpain activity decreased during storage due to the consequence of higher levels of cross-linked proteins as well as protein oxidation [35].

The reduction in hardness and increase in liquid loss in all groups by elapsing storage time was probably due to the enzymatic degradation of proteins reported before [36,37]. The obtained results indicated the most significant reduction in hardness during the first three days of storage in all samples owing to the calpain activity as the main initiator of fillet softening followed by oxidation development [37,38]. Moreover, higher liquid loss after 3 days storage at 120 kg·m^−3^ in bled compared to unbled might be related to a higher protein oxidation rate.

Furthermore, concerning the bleeding, we observed a higher hardness and lower liquid loss in bled samples compared to unbled samples due to the lower oxidation progress in bled fillets. However, hardness increased on the last day of storage in unbled fish, revealing the different pathways of oxidation development. Bleeding prevented extensive oxidation, which probably led to protein aggregation and higher hardness on the last day of storage, while moderate oxidation decreased hardness [6].

The stock density in the current study showed a better hardness at 90, and 120 kg·m^−3^ in both bled and unbled fillets compared to 150 kg·m^−3^, which confirmed the role of fish welfare during culture on fillet quality. In contrast, [39] did not report any significant differences in hardness among all stock densities. Ref. [40] suggested that a firmer texture results from more rigid connective tissue caused by exercise. We proposed that fish at 90 and 120 (kg·m^−3^) stock densities might have more opportunity for intense swimming, positively correlated with a better texture. In this regard, we revealed almost the same textural quality in bled fillets at either 90 or 120 (kg·m^−3^) stock densities, which would recommend 120 (kg·m^−3^) stock density for African catfish production. Thus, higher stock density is in contrast with fish welfare and showed negative impact on fillet quality.

We observed a reduction in Hb content, which has been examined and reported before in Asian seabass (*Lates calcarifer*) and common carp (*Cyprinus carpio*) [7,8]. The breakdown of heme proteins could likely be considered the primary possibility for heme reduction through the storage period [40]. Additionally, protein degradation in both bled, and unbled groups might be a supplementary pathway to decrease Hb [41].

We detected a significantly higher lipid oxidation progress in bled fillets than in unbled fillets during 9-day refrigerator storage. This may be due to the role of Hb in accelerating lipid oxidation [42], as suggested by earlier studies by [7,8]. Ref. [43] indicated lipid oxidation development in minced trout muscle was retarded by bleeding during storage at +2 °C.

In the current study, bleeding removed approximately 45% of blood from bled muscles. The consequence of bleeding is retarding lipid oxidation development, which is confirmed by the amount of MDA in bled fish fillets [43]. Additionally, the higher pH in bled fillets compared to unbled fillets would be an alternative reason to reduce the formation of met-Hb and Hb-deoxygenation and minimize lipid oxidation progress. Moreover, the higher protein carbonyl content in unbled fillets might be related to the higher protein oxidation progress. The results showed the importance of stock density on oxidation progress. The results of this study indicated the consequences and importance of adequate fish bleeding to retain a better fillet quality. A higher amount of Hb in both bled and unbled fish at either 90 and 120 stock densities compared to 150 (kg·m^−3^) might be related to the lower fish welfare condition and oxygen level in the water. Our results revealed the highest lipid and protein oxidation rate in the first 3 days of storage in unbled fillets compared to bled fillets. This was most likely due to the higher amount of Hb in the unbled fillet, while after 3 days, the oxidation rate was reduced due to Hb degradation. The obtained results in unbled fillets in the current study revealed that after 3 days, the oxidation rate dropped from 240% to 50%, but the reduction rate of Hb content was stable (20%). These observations consequently indicated the different Hb formation in unbled fillets that could change the lipid oxidation rate in this study. For instance, it is believed that deoxyHb can stimulate lipid oxidation more than metHb and oxyHb [44]. Therefore, lipid oxidation development and the percentage of deoxyHb in fish muscle are closely related [45]. Consequently, it is plausible that deoxyHb was dominant in unbled fillets for the first 3 days of storage, while by elapsing the time, metHb and oxyHb were dominant.

SDS–PAGE analysis was used to identify the protein pattern alterations in all groups over the time of refrigerator storage. More weak protein bands were predominantly identified in unbled fillets than in bled fillets. However, the band intensity at approximately 220 kDa decreased very pronounced in all groups, especially in the unbled group, by elapsed time. This result is in accordance with previous studies [33,46], which reported a decreasing trend in MHC band intensity. Apparently, MHC was not resistant to proteolysis in this study probably due to the formation of cross-linked proteins [47].

Additionally, the current study revealed more protein carbonyls in unbled fillets than in bled fillets during storage, which was confirmed by spectrophotometric analysis and Western blotting. More intense protein carbonyls in unbled fillets compared to bled fillets might be a consequence of protein oxidation mainly on MLC. Protein oxidation through metal-catalyzed cleavages can be considered a main pathway of oxidative damage during postmortem conditions [6]. Most likely, a higher level of hemoglobin leads to higher protein oxidation (via metal-catalyzed) in unbled fillets. Likewise, better textural parameters (WHC and hardness) in bled fillets compared to unbled fillets can be related to the less oxidized protein. Western blotting with respect to the stock density did not show any difference among all the groups, while bled fillets from 150 kg·m^−3^ density suggests more oxidized protein carbonyls compared to 90 and 120 kg·m^−3^ due to the higher amount of Hb. As touched upon above, lower hardness in bled fillets from fish reared at 150 kg·m^−3^ is mostly related to more oxidized protein compared to 90 and 120 kg·m^−3^. According to the obtained results, the authors accordingly suggested that in unbled fillets, lipid oxidation and protein degradation are started together, while in bled fillets, protein degradation is started earlier than lipid oxidation is dominated, which was confirmed by Western blot.

In turn, further in-depth studies are needed to understand better the effects of stocking density and the presence or absence of bleeding at slaughter on the different variables measured in this study.

## 5. Conclusions

The current study indicated the consequences of fish welfare on African catfish fillet quality during postmortem storage with respect to different analysis. Both of the considerations regarding fish welfare (bleeding and stock density) showed a remarkable influence on the final fillet quality. A higher oxidation progress mainly in the first 3 days of storage in unbled fillets compared to bled fillets might be related to the different formations of hemoglobin. Therefore, we proposed that the first three days of storage in the most crucial time to retard the oxidation progress. Hardness analysis revealed two different protein oxidation pathways (extensive and moderate protein oxidation) in unbled and bled fillets, respectively, in the last day of storage. Bled fillets showed a better textural parameter and lower microbial growth than unbled fillets. Western blot analysis showed more intense bands in unbled fillets than in bled fillets. The stock density revealed lower hardness in fish fillets from 150 kg·m^−3^ in both the bled and unbled groups. According to the results of this study, African catfish reared in at 120 kg·m^−3^ stock density would suggest production that is two or three times lower than the currently commonly used final density of African catfish in practice. Therefore, reducing the number of fish in the rearing tank might be the next step for increasing the quality of fish fillet as well as its production. Additionally, considering postmortem changes in mitochondrial structure and function would be great to reveal what is happening during postmortem conditions. We believe that the results obtained in the current study will help reconcile the welfare of farmed fish production and fillet quality, both being important for the fishery industry.

## Figures and Tables

**Figure 1 foods-11-04090-f001:**
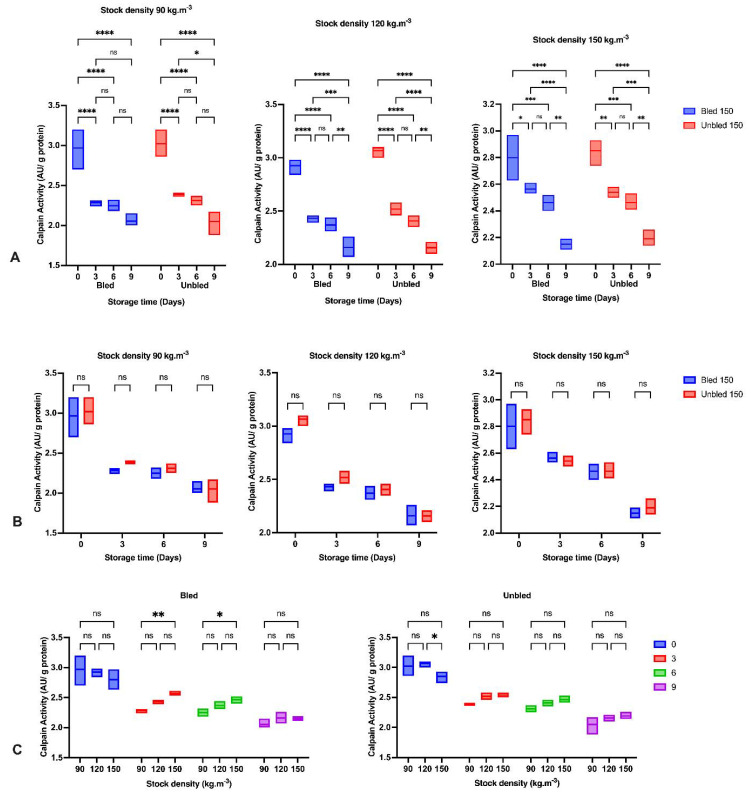
Calpain activity: (**A**) During the storage time (0, 3, 6 and 9 Days) in bled and un-bled groups at different stock densities (90, 120 and 150 kg·m^−3^); (**B**) In bled and un-bled groups at each storage time (0, 3, 6 and 9 Days) and at different stock densities (90, 120 and 150 kg·m^−3^); (**C**) In different stocking densities (90, 120 and 150 kg·m^−3^) during storage time (0, 3, 6 and 9 Days) in bled and un-bled groups; (*p* < 0.05, Tukey’s multiple comparisons test; ns: non-significant; * *p* < 0.05; ** *p* < 0.01; *** *p* < 0.001; **** *p* < 0.0001). Box plots showing line at mean. Box plots showing the line at upper quartile, lower quartile and at the mean.

**Figure 2 foods-11-04090-f002:**
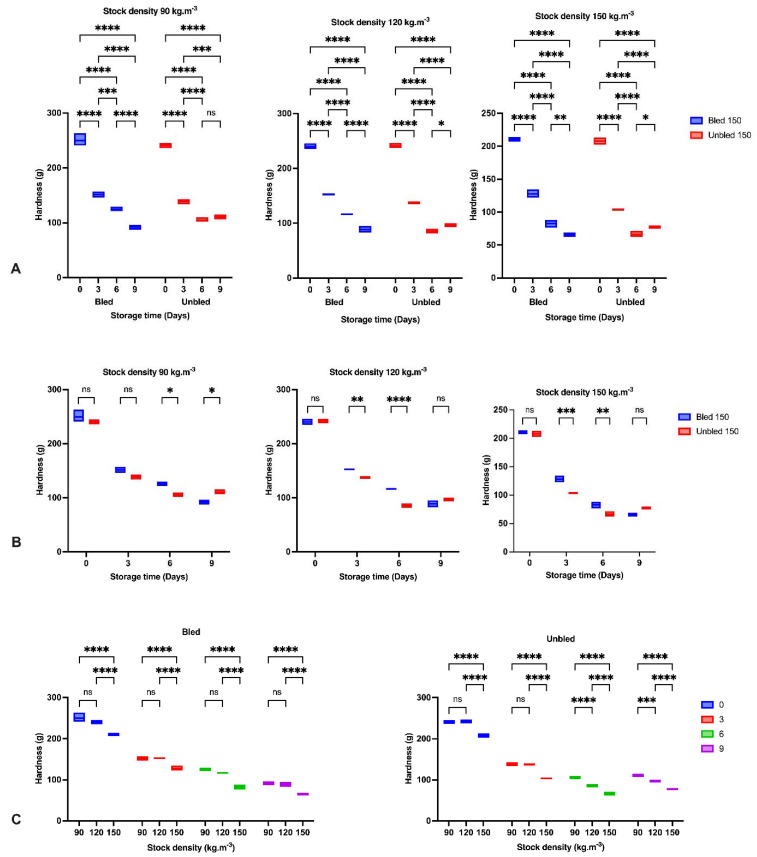
Hardness: (**A**) During the storage time (0, 3, 6 and 9 Days) in bled and un-bled groups at different stock densities (90, 120 and 150 kg·m^−3^); (**B**) In bled and un-bled groups at each storage time (0, 3, 6 and 9 Days) and at different stock densities (90, 120 and 150 kg·m^−3^); (**C**) In different stocking densities (90, 120 and 150 kg·m^−3^) during storage time (0, 3, 6 and 9 Days) in bled and un-bled groups; (*p* < 0.05, Tukey’s multiple comparisons test; ns: non-significant; * *p* < 0.05; ** *p* < 0.01; *** *p* < 0.001; **** *p* < 0.0001). Box plots showing line at mean. Box plots showing the line at upper quartile, lower quartile and at the mean.

**Figure 3 foods-11-04090-f003:**
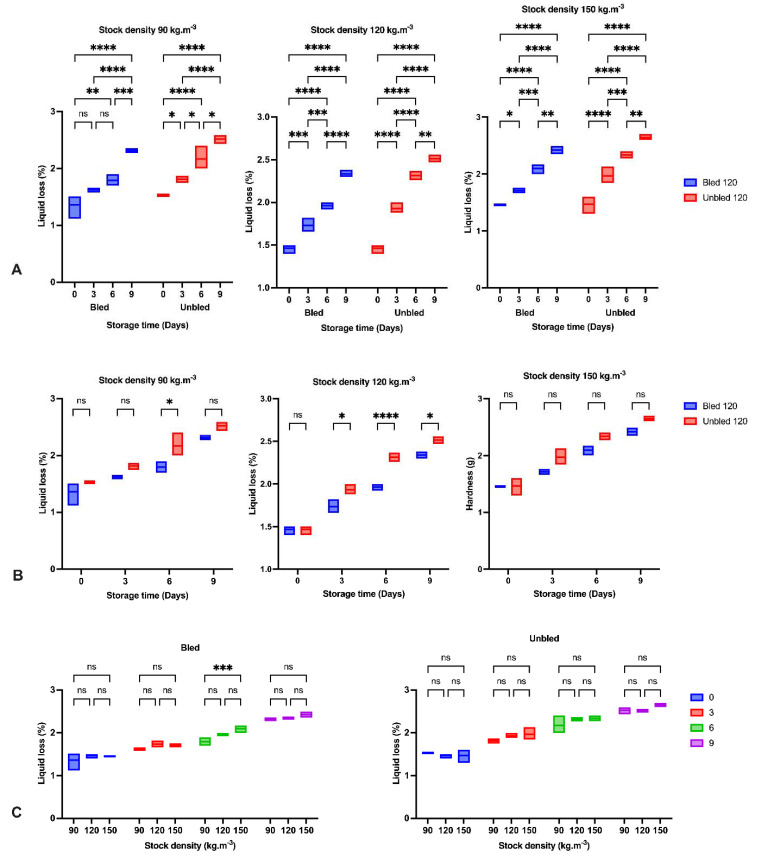
Liquid loss percentage: (**A**) During the storage time (0, 3, 6 and 9 days) in bled and unbled groups at different stock densities (90, 120 and 150 kg·m^−3^); (**B**) In bled and unbled groups at each storage time (0, 3, 6 and 9 days) and at different stock densities (90, 120 and 150 kg·m^−3^); (**C**) In different stocking densities (90, 120 and 150 kg·m^−3^) during storage time (0, 3, 6 and 9 days) in bled and unbled groups; (*p* < 0.05, Tukey’s multiple comparisons test; ns: non-significant; * *p* < 0.05; ** *p* < 0.01; *** *p* < 0.001; **** *p* < 0.0001). Box plots showing line at mean. Box plots showing the line at upper quartile, lower quartile and at the mean.

**Figure 4 foods-11-04090-f004:**
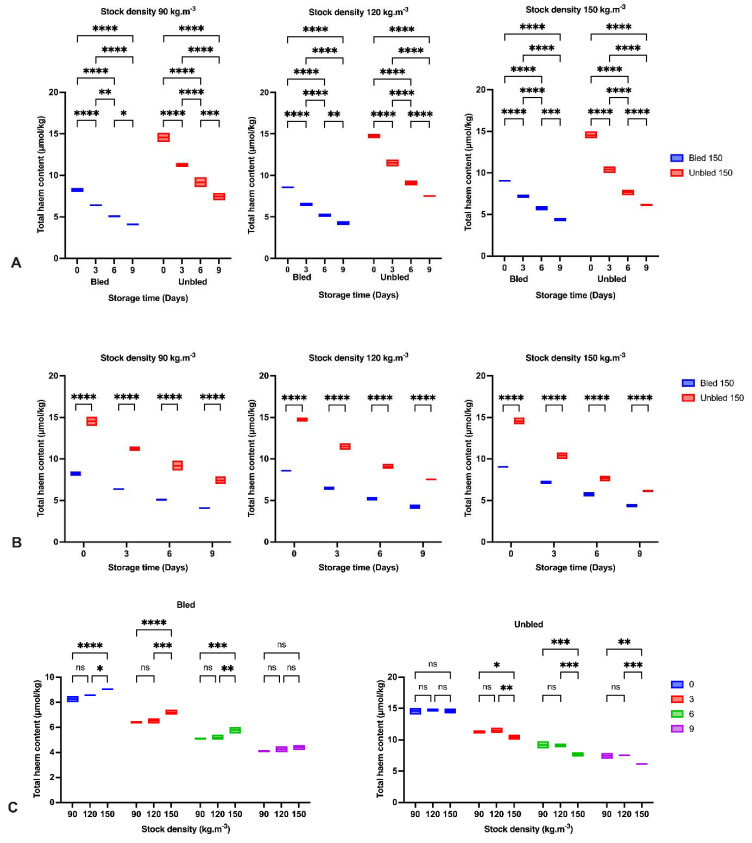
Total heme content: (**A**) During the storage time (0, 3, 6 and 9 days) in bled and unbled groups at different stock densities (90, 120 and 150 kg·m^−3^); (**B**) In bled and unbled groups at each storage time (0, 3, 6 and 9 days) and at different stock densities (90, 120 and 150 kg·m^−3^); (**C**) In different stocking densities (90, 120 and 150 kg·m^−3^) during storage time (0, 3, 6 and 9 days) in bled and unbled groups; (*p* < 0.05, Tukey’s multiple comparisons test; ns: non-significant; * *p* < 0.05; ** *p* < 0.01; *** *p* < 0.001; **** *p* < 0.0001). Box plots showing line at mean. Box plots showing the line at upper quartile, lower quartile and at the mean.

**Figure 5 foods-11-04090-f005:**
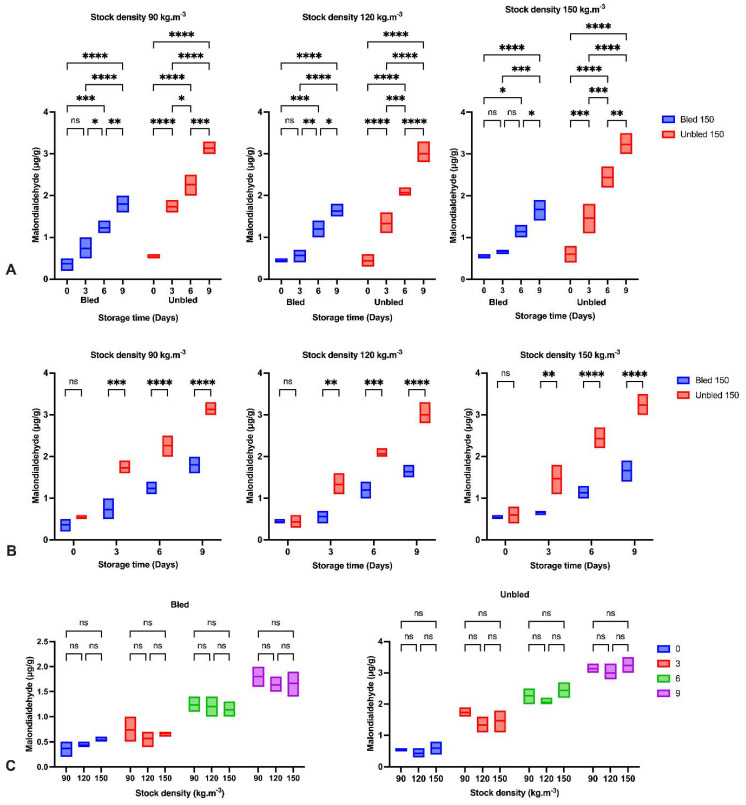
TBARS: (**A**) During the storage time (0, 3, 6 and 9 days) in bled and unbled groups at different stock densities (90, 120 and 150 kg·m^−3^); (**B**) In bled and unbled groups at each storage time (0, 3, 6 and 9 days) and at different stock densities (90, 120 and 150 kg·m^−3^); (**C**) In different stocking densities (90, 120 and 150 kg·m^−3^) during storage time (0, 3, 6 and 9 days) in bled and unbled groups; (*p* < 0.05, Tukey’s multiple comparisons test; ns: non-significant; * *p* < 0.05; ** *p* < 0.01; *** *p* < 0.001; **** *p* < 0.0001). Box plots showing line at mean. Box plots showing the line at upper quartile, lower quartile and at the mean.

**Figure 6 foods-11-04090-f006:**
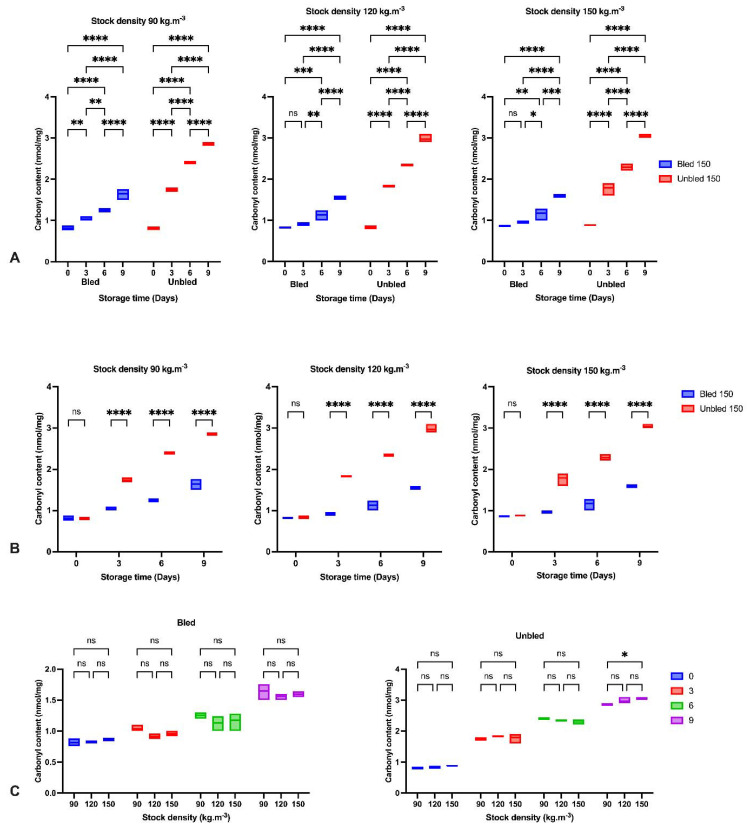
Carbonyl content: (**A**) During the storage time (0, 3, 6 and 9 days) in bled and unbled groups at different stock densities (90, 120 and 150 kg·m^−3^); (**B**) In bled and unbled groups at each storage time (0, 3, 6 and 9 days) and at different stock densities (90, 120 and 150 kg·m^−3^); (**C**) In different stocking densities (90, 120 and 150 kg·m^−3^) during storage time (0, 3, 6 and 9 days) in bled and unbled groups; (*p* < 0.05, Tukey’s multiple comparisons test; ns: non-significant; * *p* < 0.05; ** *p* < 0.01; *** *p* < 0.001; **** *p* < 0.0001). Box plots showing line at mean. Box plots showing the line at upper quartile, lower quartile and at the mean.

**Figure 7 foods-11-04090-f007:**
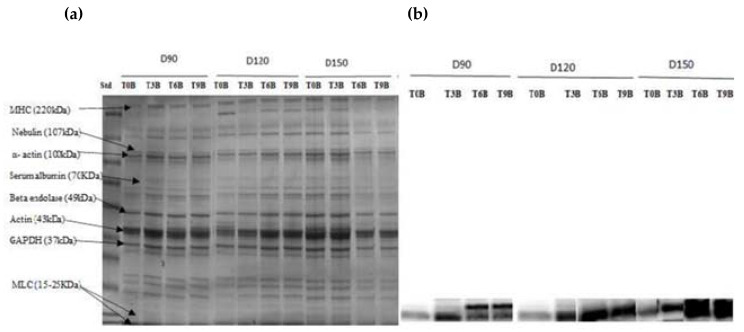
(**a**) SDS-polyacrylamide gel electrophoresis, (**b**) immunoblotting against protein carbonyl groups in bled African catfish fillet from three different stock densities during refrigerated storage at +4 °C. T: Time, B: bled, D: density.

**Figure 8 foods-11-04090-f008:**
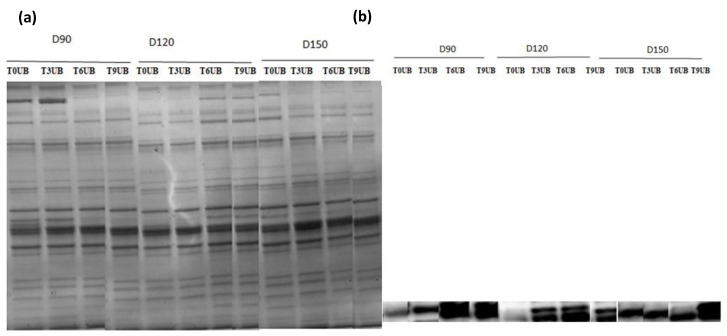
(**a**) SDS-polyacrylamide gel electrophoresis, (**b**) immunoblotting against protein carbonyl groups in bled African catfish fillet from three different stock densities during refrigerated storage at +4 °C. T: Time, B: bled, D: density.

**Table 1 foods-11-04090-t001:** Changes in the total viable count (TVC) and psychrophilic bacterial count (PBC), in bled and unbled African catfish during fridge storage.

	TVC (Log10 Cfu g^−1^)		PBC (Log10 Cfu g^−1^)	
Time (Days)	Bled	Unbled	Bled	Unbled
0	2.2 ± 0.02 ^Aa^	2.3 ± 0.01 ^Aa^	2.5 ± 0.01 ^Aa^	2.5 ± 0.03 ^Aa^
3	2.8 ± 0.01 ^Ba^	3 ± 0.02 ^Ba^	3 ± 0.02 ^Ba^	3.6 ± 0.01 ^Bb^
6	3.5 ± 0.03 ^Ca^	4.3 ± 0.01 ^Cb^	4.2 ± 0.02 ^Ca^	5.3 ± 0.02 ^Cb^
9	5.4 ± 0.02 ^Da^	6.7 ± 0.01 ^Db^	6.6 ± 0.01 ^Da^	7.4 ± 0.01 ^Db^

Different capital letters in the columns indicate significant differences (*p* < 0.05) in each group (bled and unbled) by elapsing time. Small letters in the columns indicate significant difference (*p* < 0.05) between bled and unbled fillets at the same time point. (mean ± S.D., *n* = 5).

## Data Availability

The data presented in this study are available on request from the corresponding author. The data are not publicly available due to the sensitive nature of the research.

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
