# Peer review of "Considering Two Aspects of Fish Welfare on African Catfish (Clarias gariepinus) Fillet throughout Postmortem Condition: Efficiency and Mechanisms"

_foods, 2022, doi:10.3390/foods11244090_

Round 1
Reviewer 1 Report
The paper is well written. I think that the topic is relevant and practical. Please find some comments in the text and do minor revision.

Author Response
Point 1: Page 1, Lines 33.
Response 1: We included a new reference in line 53.
Point 2: Page 2, Lines 81.
Response 2: Yes, the number of fish was the same in all groups with respect to the density and slaughtering methods. We included this information in line 116-117.
Point 3: Table 1:
Response 3: Thank you for your comment. We uniformed and corrected the table accordingly.

Reviewer 2 Report
General comments:
This is a very timely and interesting paper given that the topic of fish welfare and environmental enrichment have got increasing attention from diverse fields. Overall, I found the paper to be excellent and the end results and conclusions are not very surprising. The introduction and result sections are easy to understand and look logical. The method section is thorough but may still need to add some details. The conclusion may need some revisions.
Some minor observations:
1) I suggest the authors upload their raw data as supplementary data. This information is important for readers who want to conduct further research (e.g., meta-analysis) in the future.
2) L57-58: The authors seem to miss some important recent publications (e.g., Kristiansen et al., 2020; Zhang et al., 2022). Adding these papers to the reference list will strongly increase the persuasion of the present manuscript. In addition, in my opinion, it may be more proper to elaborate the content of fish welfare. Fish welfare has abundant connotation. Briefly describe its connotation and category may make the paper attract more readers. I suggest the authors consider this point.
Kristiansen, T.S., Fernö, A., Pavlidis, M.A., van de Vis, H., 2020. The welfare of fish, Springer, Cham.
Zhang, Z., Gao, L., Zhang, X., 2022. Environmental enrichment increases aquatic animal welfare: A systematic review and meta-analysis. Rev. Aquacult. 14(3), 1120-1135.
3) L64-68: Please add your predictions.
4) L73: Please add the license code.
5) L142-143: Please add the reference.
6) L234: Did you test the tank effect? Because you have three tanks each treatment, including tank as a random factor into the models may be more proper.
7) L243: Did you test the normality and homogeneity of your data before ANOVA?
8) L304: Add the information of meanings of upper and lower border of the box. The other figures are similar.
9) L526: Please briefly describe the drawbacks of the present study and the potential future directions.
Author Response
Point 1: I suggest the authors upload their raw data as supplementary data. This information is important for readers who want to conduct further research (e.g., meta-analysis) in the future.
Response 1: We provided the raw data and uploaded as a supplementary data.
Point 2: L57-58: The authors seem to miss some important recent publications (e.g., Kristiansen et al., 2020; Zhang et al., 2022). Adding these papers to the reference list will strongly increase the persuasion of the present manuscript. In addition, in my opinion, it may be more proper to elaborate the content of fish welfare. Fish welfare has abundant connotation. Briefly describe its connotation and category may make the paper attract more readers. I suggest the authors consider this point.
Response 2: Thank you for your suggestions. We included the references and some information about fish welfare in lines: 83-87.
Point 3: L64-68: Please add your predictions.
Response 3: Thank you for your comment. We included this information in lines: 97-102.
Point 4: L73: Please add the license code
Response 4: We added the code in line: 107-108.
Point 5: L142-143: Please add the reference
Response 5: That is correct we included in line 179. We already published some papers according to this method.
Point 6: L234: Did you test the tank effect? Because you have three tanks each treatment, including tank as a random factor into the models may be more proper.
Response 6: We appreciate your comment. We would like to inform you that we kept all fish in the same tank size, color, and water condition to minimize the tank affect. Therefore, we believe that this affect might be the same for all groups which was not the aim of this study.
Point 7: L243: Did you test the normality and homogeneity of your data before ANOVA?
Response 7: Yes, we did the normality and homogeneity step before the ANOVA analysis. We included this information in lines: 288-289.
Point 8: L304: Add the information of meanings of upper and lower border of the box. The other figures are similar
Response 8: Yes, we corrected and included the description of box plot in all the figures “Box plots showing the line at upper quartile, lower quartile and at the mean.”
Point 9: L526: Please briefly describe the drawbacks of the present study and the potential future directions.
Response 9: Thank you for your constructive comment. We included this information in lines: 665-670.

Reviewer 3 Report
This was studied to the impact of two different aspects of fish welfare (slaughtering method: bled and unbled fish; different levels of fish stock densities) on fillet quality during postmortem conditions. According to the paper, there are some studies about sea fish have reported the similar papers, please explain the innovativeness. The other, the language needs to be improved.
1. Title need to be improve.
2. Abstract is also not reflect the content of the article and needs to be rewritten.
3. L48-56 the sentences need to rewritten.
4. The aim of this paper is not clear.
5. More information of Raw materials are not also provided.
6. L 101 What about the reference of Bito, 1983
7. L 154 What about the reference of Gomez-Ba-sauri & Regenstein, 1992)
8. ml or mL?
9. 2.6. Hardness analysis need to rewritten. Because the
10. Statistical Analysis need to be rewritten.
11. Discussion need to be rewritten.
12. Figures and Tables need to be rewritten.
13. Conclusions need rewrite.
14. References need improve according to the Journal.
Author Response
Point 1: This was studied to the impact of two different aspects of fish welfare (slaughtering method: bled and unbled fish; different levels of fish stock densities) on fillet quality during postmortem conditions. According to the paper, there are some studies about sea fish have reported the similar papers, please explain the innovativeness. The other, the language needs to be improved.
Response 1: Thank you for your constructive comment. We assessed a comprehensive study on the African catfish fillet during postmortem influenced by fish welfare. In this study we carefully considered two important aspect of fish welfare and tried to find a relationship between all the indicated parameters. Additionally, the impact of fish welfare on protein was investigated by proteomics approach. We also, suggested the mechanisms oxidation pathway in the current study. We would like to inform you that we sent the manuscript to the Elsevier Language Editing service accordingly with the reference number LE316321.
Point 2: 1. Title need to be improve.
Response 2: We changed the title accordingly.
Point 3: Abstract is also not reflect the content of the article and needs to be rewritten.
Response 3: We rewrote abstract accordingly. Please find in lines 15-28.
Point 4: 3. L48-56 the sentences need to rewritten.
Response 4: We rewrote these sentences accordingly in lines: 68-74.
Point 5: The aim of this paper is not clear.
Response 5: We appreciated your comments and corrected this part in lines: 97-102.
Point 6: More information of Raw materials are not also provided.
Response 6: That is right. We included this information in lines 44-49.
Point 7: L 101 What about the reference of Bito, 1983
Response 7: That is right. We corrected in line 139.
Point 8: L 154 What about the reference of Gomez-Ba-sauri & Regenstein, 1992)
Response 8: Yes, we corrected in line 192.
Point 9: ml or mL?
Response 9: Yes, we corrected in line 194.
Point 10: 2.6. Hardness analysis need to rewritten. Because the
Response 10: Thank you for your comment. We rewrote and corrected this part accordingly in lines: 175-181.
Point 11: Statistical Analysis need to be rewritten.
Response 11: We rewrote this part.
Point 12: Discussion need to be rewritten.
Response 12: Thank you for your constructive comment. We rewrote this part accordingly.
Point 13: Figures and Tables need to be rewritten.
Response 13: We appreciate your comment. We included new caption part for each graph and corrected the table accordingly.
Point 14: Conclusions need rewrite.
Response 14: Thank you for your constructive comment. We rewrote this part accordingly.
Point 15: References need improve according to the Journal.
Response 15: we agree and corrected according to the journal format.

Reviewer 4 Report
The manuscript studies the impact of two different aspects of fish welfare, slaughter method, and stocking densities, on fillet quality during refrigerated storage.
The assay presented is complex and different effects are evaluated. Then it is necessary to write the statistical analysis in more detail. The effects considered, their interactions, the experimental unit, and the model used must be pointed out.
How were tanks considered in the model?
If the effects were bled/unbled, storage time, and density, the significance of the double and triple interactions should be analyzed.
In addition, it is requested to indicate the statistical significance when describing the results in section 3, for effects and interactions. This will allow a better understanding of the results since they are presented in a disaggregated mode.
Line 92-93
“...six random fillets were used for chemical analyzes (Supplementary Materials Table S1).”
What chemical analysis are you referring to?
Line 45 “re-sults” correct
Line 115 2.4. Total viable counts and psychrophilic bacteria counts
In this analysis, were storage time and density considered?
Lines 424 - 437
Please explain these sentences.
“TVC indicated that the bled fillets were of good quality and that the unbled fillets were in an acceptable condition.”
“The presence of blood in the unbled fillets may provide a suitable substrate for the growth of microorganisms, evidenced by the higher amount of TVC and PBC during the storage time of 9 days [7]. Therefore, regarding the microbiological point, we suggest 6 and 9 days of shelf life for bled and unbled African catfish fillets during refrigerated storage.”
Author Response
Point 1: The assay presented is complex and different effects are evaluated. Then it is necessary to write the statistical analysis in more detail. The effects considered, their interactions, the experimental unit, and the model used must be pointed out.
How were tanks considered in the model?
If the effects were bled/unbled, storage time, and density, the significance of the double and triple interactions should be analyzed.
In addition, it is requested to indicate the statistical significance when describing the results in section 3, for effects and interactions. This will allow a better understanding of the results since they are presented in a disaggregated mode.
Response 1: We studied multi-interactions between different variabales. Bled and Unbled, with Density at 90, 120 and 150 kg.m-3 and Storage time: 0, 3, 6 and 9 days. Therefore, we did the two-way ANOVA and Tukey’s multiple comparisons test to analyse the storage of fillets (0, 3, 6 and 9 days) for Bled and Unbled groups at different densities (90, 120 and 150 kg.m-3). To understand the individual effect of culture densities (90, 120 and 150 kg.m-3) for bled and unbled conditions, we needed to analyse the data separately by two-way ANOVA. As our comparisons were Bled and Unbled, Density at 90, 120 and 150 kg.m-3 and Storage time: 0, 3, 6 and 9 days (2 X 3 X4 comparisons) at the same time, therefore the three-way ANOVA wasn’t possible to perform on our data. However, we tried to present our complex data in a most straightforward form to make it easier to understand.
Point 2: Line 92-93
“...six random fillets were used for chemical analyzes (Supplementary Materials Table S1).”
What chemical analysis are you referring to?
Response 2: We included the analysis in line: 130-131.
Point 3: Line 45 “re-sults” correct
Response 3: We corrected in line 290.
Point 4: Line 115 2.4. Total viable counts and psychrophilic bacteria counts. In this analysis, were storage time and density considered?
Response 4: We appreciate your comments about this issue. We considered the microbial growths by elapsing time (as shown in table 1), but we did not consider the impact of density. We would not expect to see any influence of density on microbial growth.
Point 5: Lines 424 - 437
Please explain these sentences.
“TVC indicated that the bled fillets were of good quality and that the unbled fillets were in an acceptable condition.”
“The presence of blood in the unbled fillets may provide a suitable substrate for the growth of microorganisms, evidenced by the higher amount of TVC and PBC during the storage time of 9 days [7]. Therefore, regarding the microbiological point, we suggest 6 and 9 days of shelf life for bled and unbled African catfish fillets during refrigerated storage.”
Response 5: Thank you for your comments.
We explained both sentences in lines 508-509 and 518.
Regarding the first sentence “TVC indicated that the bled fillets were of good quality and that the unbled fillets were in an acceptable condition.” We would like to mention that the bled fillet was safe, and they showed also good quality with respect to the microbial growth. However, in unbled fillets they were in acceptable condition means that fillets were safe to consume but they were close to be spoiled.
In case of second sentence “The presence of blood in the unbled fillets may provide a suitable substrate for the growth of microorganisms, evidenced by the higher amount of TVC and PBC during the storage time of 9 days [7]. Therefore, regarding the microbiological point, we suggest 6 and 9 days of shelf life for bled and unbled African catfish fillets during refrigerated storage.”
We described that presence of blood could accelerate the microbial growth due to the suitable substrate. In the second part we corrected the mistake about the shelf life of bled and unbled fillets.

Round 2
Reviewer 2 Report
Thanks for the authors' responses.
The authors do not test the tank effect in this version, but I still think that it is very necessary to get valid results. This is my only worry for this manuscript. I do not know why authors insist not to include tank as a random factor to run their models.
Even so, the above point is a small problem and maybe does not affect their results. So I agree to accept this paper. (However, I strongly suggest authors rerun their models including the tank as a random factor).
Congratulations.
Author Response
Point 1: Thanks for the authors' responses.
The authors do not test the tank effect in this version, but I still think that it is very necessary to get valid results. This is my only worry for this manuscript. I do not know why authors insist not to include tank as a random factor to run their models.
Even so, the above point is a small problem and maybe does not affect their results. So I agree to accept this paper. (However, I strongly suggest authors rerun their models including the tank as a random factor).
Congratulations.
Response 1: We appreciate your suggestion on this issue. We would consider it in our future study.

Reviewer 3 Report
Table 1 need improve.
Author Response
Point 1: Table 1 need improve.
Response 1: Thank you for your comment. We improved table 1.

Reviewer 4 Report
The changes introduced by the authors have generated a substantial improvement in the manuscript.
Author Response
Point 1: The changes introduced by the authors have generated a substantial improvement in the manuscript.
Response 1: Thank you for your comments on our manuscript.
